# Effect of the online module on leadership in knowledge acquisition among nursing students: A randomized controlled study protocol

Jucielly Ferreira da Fonseca[1], Silmara de Oliveira Silva[1],
Louise Constancia de Melo Alves Silva[1], Roberta Paolli de Paiva Oliveira Arruda Camara[1],
Francisco de Cássio de Oliveira Mendes[1], Laura Lima Souza[2],
Rodrigo Assis Neves Dantas[1,3], Daniele Vieira Dantas[1,3*¤]

**1** Post-graduate Program in Nursing, Universidade Federal do Rio Grande do Norte, Natal, Rio Grande do Norte, Brazil, **2** Graduate, Nursing Departament, Universidade Federal do Rio Grande do Norte, Natal, Rio Grande do Norte, Brazil, **3** Nursing Department, Universidade Federal do Rio Grande do Norte, Natal, Rio Grande do Norte, Brazil

¤ Current address: Campos Universitário, Br-101, s/n - Lagoa Nova, Natal - RN. Nursing Department, Universidade Federal do Rio Grande do Norte, Natal, RN, Brazil.
* daniele.vieira@ufrn.br

## Abstract

Leadership is an essential skill for success in nursing practice, influencing the nursing team's organization, the quality of health services and decision-making in complex situations. The COVID-19 pandemic has further highlighted the need for leadership in nursing, with nurses playing a crucial role in addressing the crisis. However, there are challenges in leadership training, with conventional expository class methods often neglecting this skill. This study proposes to evaluate the impact of an online educational module focused on leadership for Nursing students, recognizing the importance of training future professionals from the beginning of their careers. To present a study protocol to evaluate the effect of an online educational module in comparison with a conventional expository class related to knowledge acquisition by Nursing students on the topic of Leadership. A Randomized Controlled Study, single-blind, two-arm will be conducted. Sixth semester students regularly enrolled in the nursing course will take part in the study. Participants will be randomly distributed into two groups, namely the Control Group, which will undergo a conventional class, and the Experimental Group, which will undergo an online educational module. Two important moments will be considered to evaluate the results of the intervention. A pesquisa, em fase de design, terá resultados, conclusões e análises definidos após a coleta de dados. The primary outcome is expected to demonstrate that the online module is equal to or superior to the conventional module in the acquisition of knowledge about leadership, addressing topics such as team involvement, organizational culture, conflict management and feedback. The secondary outcome will assess the acquisition of knowledge about leadership through an adaptation of the Nursing Student Self-Perception Questionnaire in the Exercise of Leadership.

**Data availability statement:** No datasets were generated or analysed during the current study. All relevant data from this study will be made available upon study completion.

**Funding:** The author(s) received no specific funding for this work.

**Competing interests:** The authors have declared that no competing interests exist.

## Introduction

Nurses assume a wide range of responsibilities in the context of healthcare services, from direct patient care to administrative, management, educational and research tasks. They are tasked with managing patient care and coordinating healthcare facility operations. Leadership is fundamental in all of these activities, being recognized as an essential skill for successful nursing practice [1].

Leadership in Nursing is a dynamic skill which can be cultivated and strengthened over time. Studies such as the one by Varanda *et al.* (2023) [2] highlight the direct relationship between leadership and team organization, highlighting its importance for developing quality health systems and improving patient care.

The act of leading accompanies nurses in any of the fields in which they work. Nurses are challenged to deal with complex situations, analyze information and make assertive decisions to ensure the safety and well-being of patients, as well as their team. The ability to lead is essential to face the constantly changing challenges in the healthcare environment and provide quality care [3].

Faced with the real need to encourage development of leadership in Nursing, the World Health Organization (WHO) in partnership with the International Council of Nurses (ICN) launched the "Nursing Now Challenge" in 2018. The Program was strengthened in Brazil through the partnership between the Federal Nursing Council (*Conselho Federal de Enfermagem - COFEN*) and the OMS/OPAS, with one of the campaign's primary goals being to strengthen the education and development of Nursing professionals with an emphasis on leadership [4].

Nursing leadership during the COVID-19 pandemic was tested on different fronts, requiring specific skills to face unprecedented challenges. As the largest professional category in healthcare, nurses have not only assumed clinical responsibilities, but also led teams, managed resources and provided emotional support amidst a scenario of great stress and uncertainty. Thus, leadership has emerged as an essential element in maintaining the cohesion and effectiveness of healthcare teams during this challenging period [5].

There are still challenges in implementing leadership teaching in the training of nurses 22 years after the promulgation of the Curricular Guidelines for the Undergraduate Nursing Course. This occurs due to the outdated use of teaching methodologies, the fragility of teaching related to the lack of updating of teachers regarding teaching leadership and educational methods which have roots in conventional and traditional education [6].

As a way of mitigating the deficiency in teaching regarding the subject and given the possibility of exploring what digital educational technologies can provide to the learning process, the paradigm of teacher-centered education is broken and begins to value the student, providing an education capable of promoting meaningful learning [7].

Considering the above, there is a need to evaluate the effect of an online educational module in comparison with a conventional expository class related to knowledge acquisition by Nursing students on the topic of Leadership.

## Materials and methods

This protocol will adhere to the Consolidated Standards of Reporting Trials (CONSORT) [8] and standard protocol item statements for randomized trials [9]. This study was approved by the Research Ethics Committee of the Universidade Federal do Rio Grande do Norte - UFRN, under CAAE No. 75851023.0.0000.5537 and Registry: RBR-2dfqmr2 - Registro Brasileiro de Ensaios Clínicos – REBEC (26/02/2024).

### Study design

This is a Randomized Controlled Study in which participants will be selected at random and the study will be conducted in a single-blind manner, with assessments before and after the

intervention. To do so, participants will be divided into a Control Group, which consists of subjects whose performance in relation to a dependent variable is used as a parameter to evaluate behavior; and the Experimental Group is the group of participants who receive the intervention [10]. Participants in the Control Group will undergo a conventional class while the other participants in the Experimental Group will undergo an online educational module.

## Study scenario

Recruitment and intervention will be conducted at the Nursing Department of the Federal University of Rio Grande do Norte (UFRN), Campus Natal/RN, Brazil, with undergraduate Nursing students from this institution.

## Study population

The population of this study will consist of sixth semester undergraduate nursing students from the Federal University of Rio Grande do Norte. The sample size will be one of convenience and will include only one class, with an average of 40 participants, who will be randomly divided between the control group and the experimental group.

As the university where the study will take place only offers one seventh semester class per academic year, it is not possible to exceed the number of students in the sample established in this protocol. We therefore opted for a convenience sample, which represents a limitation of the study, since it is not possible to guarantee the generalizability of the results. However, this protocol could be reproduced in future studies with larger samples, which would allow for greater coverage and potential generalization of the findings.

## Selection criteria

Person of both sexes, aged 18 or over; be a student in the sixth semester of the UFRN Nursing course. Students in the sixth semester who have already attended other degrees, *stricto sensu* or *lato sensu* postgraduate courses, extension courses, free courses, short duration courses, or events that address the topic of leadership and absentees will be excluded from the study.

## Recruitment

Sixth semester students will be recruited by offering a 30-hour "Nursing Leadership Course" extension course, registered in the institution's Integrated Academic Activities Management System (SIGAA).

Firstly, students in the sixth semester of the undergraduate nursing course at the Federal University of Rio Grande do Norte will be invited via the class Whatsapp group, encouraging them to sign up for the course.

Interested students will have to fill in a registration form available on the Google Forms platform, which will collect initial information and confirm their interest in taking part in the study. During registration, participants will agree to the Informed Consent Form (ICF), ensuring that they are aware of the stages of the study.

After registration, a Google Meet call will be scheduled with all registered students. During this call, a pre-test will be administered to assess prior knowledge about leadership. Participants who complete the pre-test and confirm their adherence to the study will be added to a WhatsApp group to receive important notices and information about the course.

In order to minimize interaction between the control and experimental groups, participants will be given clear instructions on the importance of not sharing information between the groups, with the aim of preserving the integrity of the study, since they will be part of different Whatsapp groups but will not know which group they have been allocated to.

## Allocation

To ensure random allocation into groups, simple randomization will be applied using the website www.random.org. This tool will perform the randomization completely randomly, without researcher interference, ensuring that all participants have an equal chance of being assigned to either the control group or the experimental group.

Although the participants self-selected when they enrolled, randomization was carried out after the enrolments were collected, with the aim of ensuring that everyone had the same chance of being allocated to the different experimental groups.

Participants in the control group will attend a conventional expository class, held in person at the Nursing Department. The class, conducted by the researcher, will last four hours in the afternoon and cover the same topics as those addressed in the online educational module. Both the experimental group, which will have access to the online module, and the control group, which will have access to the conventional class, will last 30 hours. The control group will attend a lecture given by the researcher, covering the same topics and in the same depth as the online module at the Nursing Department at the Federal University of Rio Grande do Norte, lasting four hours. Although the in person class will last four hours, the total workload for the control group will also be 30 hours, as there will be suggested supplementary reading to be consumed.

Meanwhile, participants in the experimental group will engage in an online educational module, with recorded classes hosted on the Google Classroom virtual learning environment. Participants will have a period of five days to access and consume the module's content, totaling 30 hours, from registration to completion. After the class, all the participants in the research, both the control group and the experimental group, will be called to go in person to the Nursing Department of the Federal University of Rio Grande do Norte to complete the post-test and receive certification of participation in the course.

It is important to emphasize that participant allocation will be conducted in a single-blind manner to ensure impartiality in the process. The sample for this study will be divided equally between the two groups, ensuring that both the Control Group and the Experimental Group have an equivalent number of participants. Additionally, participant assignment will remain confidential to prevent any external interference or bias from the researcher or third parties.

## Initial evaluation

The researcher will approach students virtually. Participants who agree to be part of the study, by signing the Informed Consent Form, will be added to a WhatsApp group where necessary notifications will be sent. After the registration is complete, participants will be introduced to the study and informed about the interventions.

Any questions related to the research will be clarified before the start of the interventions. Subsequently, participants will be assigned to one of the two groups (Control Group or Experimental Group) based on a randomization list.

## Intervention

The sample will consist of approximately 40 students from the sixth semester of the Nursing course who are regularly enrolled. Throughout their participation, both online and in person, the participants will use exclusively their own mobile devices and internet access for registration and the pre-test.

The pre-test will be administered via Google Forms, and no prior support materials will be provided. Participants will have 20 minutes to complete the pre-test, which will be conducted by the researcher using a Google Meet room as the virtual space to accommodate all

participants, as everyone needs to answer the pre-test synchronously within the estimated 20 minutes. The virtual tool's chat feature will be used to signal completion. All participants must respond within the allocated time.

Randomization will be conducted only with participants who complete the pre-test and meet the eligibility criteria. These students will be randomly assigned to either the Control Group or the Experimental Group. After randomization, participants will be informed via WhatsApp and email about which group they belong to: those attending the in-person course at the Nursing Department or those consuming the online educational module.

Participants in the Experimental Group will be enrolled in a Google Classroom course by the researcher, with instructions sent via email and WhatsApp. The online educational module, consisting of interactive recorded classes, will have a duration of four hours, with additional content such as digital materials, quizzes, and forums. Topics covered include team motivation and engagement, organizational culture, conflict management, and constructive feedback. Participants will have flexibility in accessing the classes and will not be restricted by geographical location.

The Control Group will attend a conventional expository class held in person at the Nursing Department of the Federal University of Rio Grande do Norte. The class will use visual materials, such as PowerPoint presentations, covering the same topics as the online module with equal depth and breadth.

The Experimental Group will also attend the Nursing Department in person after the Control Group's activities. Following this, both groups will complete the post-test online via Google Forms. Participants will use their own electronic devices and access the UFRN internet. They will have 20 minutes to complete the post-test. After completion, all participants will receive a certificate of participation in the course and will be released.

Thus, the pre-test will be conducted synchronously and remotely, before the intervention, using Google Forms, with participants gathered in a Google Meet room, 20 minutes and no prior support material will be provided. After the pre-test, participants will be allocated and randomized into groups. The post-test will be conducted after all planned interventions are completed, in person using Google Forms. Participants will again have 20 minutes to complete the questionnaire. The researcher will be present during the application to ensure that all participants have the same conditions of time and environment to complete the test.

The purpose of the evaluation is to assess the effectiveness of an online educational module on leadership in the knowledge acquisition of Nursing students.

## Data collection instrument

To evaluate the effectiveness of the module and the acquisition of knowledge by the participants, a form developed in Google forms will be used, applied as a pre-test and post-test for both the control group and the experimental group.

The data collection forms will consist of two documents. The first is the registration form with demographic characterization, which will collect personal and academic information relevant to the study; while the second is a form that will be used as a pre- and post-test containing a problem situation made up of four objective questions focused on topics such as team motivation and engagement, organizational culture, conflict management and constructive feedback, in addition to the adaptation of the Nursing Student Self-Perception Questionnaire on Exercising Leadership (QUAPEEL), developed by Cardoso, Ramos and D'Innocenzo (2014) [11].

The problem situation is made up of six objective questions, and each question contains four alternative answers. The questions in the document are: 'Faced with the possible

dismissal of nursing professionals due to the salary floor, how would you, as a leader, address this issue to keep the team motivated and engaged?'; 'Faced with signs of stress in the team, how would you promote an organizational culture that effectively addresses work overload and preserves the well-being of professionals? '; 'In an environment where conflicts can arise due to difficult ethical decisions, as a leader, how would you mediate a conflict between team members with opposing views on the allocation of scarce resources?'; 'As a leader, how would you provide constructive feedback to team members who have made medication errors, while promoting professional development?'; 'How do you conceptualize leadership?'; 'Do you consider yourself a leader?'; 'Tick the interpersonal skills you consider necessary for a leader?'

QUAPEEL is a questionnaire that measures the skills and attitudes that nurses should exercise in their leadership practice. For use in this study, the questionnaire will be adapted in the section referring to the sociodemographic question, since the instrument will be applied to students and not to nurses.

The application of QUAPEEL involves three main parts: the first part collects sociodemographic information about the interviewees, such as age, education and professional experience; the second part consists of questions that explore nurses' knowledge about leadership, allowing for a qualitative and quantitative assessment; and the third part focuses on specific leadership competencies, addressing skills and attitudes that nurses believe they possess or need to develop.

The third part of the questionnaire consists of 20 statements that assess the dimensions of the coaching process, with each dimension having six specific situations. For each statement, there are the following alternatives: "1. Never" - I don't understand the statement; "2. Rarely" - I understand the statement sometimes; "3. Not always" - I understand the statement sometimes; "4. Almost always" - I understand the statement many times; "5. Always" - I understand the statement all the time; "NA. Not applicable" - if you have no way of evaluating the question. The instrument's score ranges from 0 to 100, with values closer to 0 corresponding to the lowest perception and values closer to 100 to the highest perception of Leadership practice.

These instruments will allow for both quantitative and qualitative assessment, as well as addressing the skills and attitudes that students believe they possess or need to develop. This format ensures that both the control and experimental groups will have access to the same data collection instruments, promoting equality in the evaluation conditions.

Fig 1 shows the detailed stages of the study, with SPIRIT enrollment timeline, interventions and assessments.

## Blinding

The single-blind design was adopted to ensure data integrity and minimize biases. Participants will be informed on the registration form about the general differences between the two teaching modalities. However, the specific allocation of participants to each group will be conducted confidentially and randomly to avoid external interference and reduce bias. This approach guarantees transparency with the participants while maintaining the impartiality necessary for the validity of the study.

The researcher teaching the in person course for the control group will necessarily know the participants in this condition. However, to mitigate any potential bias arising from this limitation, the statistician responsible for processing the data will not know which group each participant will be in, so that the analysis of the data is not influenced during or after the study, until the final analyses have been completed. To ensure impartiality in data handling, the data were presented without any identification that would reveal to which group each participant belonged.

| STUDY PERIOD | | | | | |
|---|---|---|---|---|---|
| | **Enrolment** | **Baseline** | **Post-allocation: Allocation, Interventions and Follow-up** | | |
| **Timepoint** | - | - | **Allocation** | **Intervention** | **Follow-up** |
| **Enrolment** | | | | | |
| Eligibility criteria | X | | | | |
| Recruitment | X | | | | |
| Intial assessment | X | | | | |
| Informed consent | X | | X | | |
| Allocation | | | | | |
| **Intervention** | | | | | |
| Traditional expository class (control group) | | | | X | |
| Online educational module (intervention group) | | | | X | |
| **Assessments** | | | | | |
| Sociodemografic Data | | X | | | |
| Pre-intervention | | | | X | |
| Pos-intervention | | | | | X |

**Fig 1.  Description of the study stages.**

## Data collection and management

Researchers will ensure that participants' anonymity is protected and the data collected will remain confidential so that their identities and any type of identifying information are protected. Two copies of the Informed Consent Form will be provided, and signed by the participants, researchers and guardians.

After signing the consent form, the participant may terminate their participation in the study at any time if they wish, without any consequences. The results of this study may

be presented at meetings or publications; however, the identity of participants will not be revealed.

The data generated and analyzed in this study will be made publicly available in the repository of dissertations and theses of the Federal University of Rio Grande do Norte. The repository chosen follows open access guidelines, ensuring transparency and replicability of the results. In addition, the data will be presented in the form of a scientific article, which will be submitted to a specialized journal, with the aim of reaching a wider audience and contributing to the advancement of knowledge in the area.

## Management of losses or withdrawals

Losses will be carefully recorded, ensuring the reliability of the study. If any participant withdraws from the study, they will be removed from the sample. A logbook will be used to record information relevant to the study.

## Data analysis

The results analyzed will be related to the evaluation of the effectiveness of the online educational module in the acquisition of knowledge about Leadership among Nursing students, and this effectiveness will be measured by means of a pre-test and a post-test, using a problem situation instrument and the QUAPEEL questionnaire.

Therefore, in addition to the age and gender of the participants, the variables to be analyzed will include the responses of the control and experimental groups, obtained by applying the instrument containing the problem situation and the *QUAPEEL* questionnaire (S4).

The collected data will be stored and processed using Microsoft Office Excel, version 2020, and Statistical Package for the Social Sciences (SPSS), version 25.0. The analysis will include both descriptive and inferential statistics, with results presented in tables and figures.

Descriptive data treatment will involve analyzing relative and absolute frequencies, mean, standard deviation, and medians according to the normality of the variables. The normality of the data distribution will be checked using appropriate statistical tests, such as the Shapiro-Wilk test or the Kolmogorov-Smirnov test.

The answers will be analyzed using the content analysis technique and the use of a frequency scale, ranging from "Never" to "Always", aims to measure the participants' self-perception in relation to the frequency with which they apply leadership practices in their daily lives. This approach is essential for assessing not only the presence of such practices, but also the consistency with which they are used in different scenarios. Although the focus is on frequency, the interpretation of the data allows inferences to be made about the participants' understanding and use of leadership practices, in line with the objectives of the study.

To analyze the pre-test and post-test conditions of the study in this protocol, the tests to be used will depend on the distribution of the data. If the data is normally distributed, the Student's t-test for paired samples will be used to compare the means of the two moments and check for statistically significant differences.

On the other hand, if the data does not follow a normal distribution, the Wilcoxon test for paired samples will be used, which is a non-parametric test suitable for comparing the medians between the two moments. The aim of both tests is to assess whether the changes observed between the pre- and post-test are statistically significant.

## Ethical considerations

The present study was analyzed and approved by the Research Ethics Committee of the Federal University of Rio Grande do Norte – UFRN – CAAE – 75851023.0.0000.5537

(22/12//2023). This study was also registered in the Brazilian Clinical Trials Registry – REBEC – RBR-2dfqmr2 as a clinical trial (02/26/2024).

Students will participate on a voluntary and non-profit basis, and all stages, objectives, risks and benefits of the study will be explained. Participants may withdraw from the study at any stage without prejudice, without suffering penalties or judgments of any kind.

**Evaluation registration** Registry: RBR-2dfqmr2 - *Registro Brasileiro de Ensaios Clínicos – REBEC* (26/02/2024). This study was approved by the Research Ethics Committee of the Universidade Federal do Rio Grande do Norte - UFRN, under CAAE No. 75851023.0.0000.5537.

## Results

The research is in its design phase, and therefore the final results, conclusions and analysis will only be determined after implementation and data collection. The expected results reflect the initial hypotheses and objectives of the study, but may be adjusted depending on the development of the research and the analysis of the data obtained throughout the process.

### Primary outcome

It is hoped that by applying the problem situation instrument, the primary result of this study will be to demonstrate that the online educational module has an equal or greater effect on the acquisition of knowledge about leadership by nursing students, compared to the conventional classroom. This knowledge covers concepts related to team engagement, organizational culture, conflict management, feedback and the concept of leadership. To measure these variables, we used the QUAPEEL questionnaire, structured with objective and quantitative items that assess both conceptual and practical aspects.

### Secondary outcome

The secondary outcome refers to knowledge acquisition by students on the topic of Leadership, which will be assessed through an adaptation of the Nursing Student Self-Perception Questionnaire in the Exercise of Leadership (S4).

## Discussion

The discussion about Leadership competence has become increasingly frequent in the context of training and developing nurses [12]. With changes in educational practices involving different ways of teaching and learning, and to meet the demands of the job market, there is a need to rethink teaching and learning methods, seeking to integrate theoretical content about leadership with practical opportunities to apply these concepts.

The literature has emphasized the importance of innovative educational methods such as online modules to address gaps in conventional teaching. Recent research indicates that the use of online modules can promote significant evolution, not only in the understanding of the concept of leadership, but also in self-identification as a leader and in the development of other fundamental skills [13,14].

Online education offers a flexible and convenient environment, allowing students to engage in their training autonomously, adjusting to their individual learning needs and schedules. This approach not only favours progress at each individual's own pace, promoting more personalized and meaningful learning, but also eliminates geographical restrictions, guaranteeing equal access to content, as long as students have an electronic device and an internet connection [7].

Studies indicate that online training modules not only enhance knowledge, but also develop essential skills for effective leadership in nursing practice. In addition, cognitive

flexibility and digital literacy play a key role in enhancing the flow of learning, increasing student engagement and adaptability. These factors reinforce the potential of online modules in preparing future professionals to face the challenges of the professional context more efficiently and autonomously [15].

An online leadership course demonstrated effectiveness in developing leadership skills among nursing professionals in the Americas, with participants showing greater knowledge acquisition. These innovative educational practices allow students to develop practical and theoretical skills in a more dynamic and effective way, directly contributing to the advancement of the training of future nurses [16].

Although the online modality and conventional teaching is the central focus of this study, it is important to consider that the conditions of each approach involve differences beyond the form of content delivery. In the online module, participants will have access to video classes, e-books, podcasts, videos available on a streaming platform (Youtube), quizzes and a forum for discussion, promoting greater flexibility and autonomy in learning. The in person course, on the other hand, will offer direct interaction with the instructor through a conventional lecture with the use of a visual resource using Powerpoint.

These variables should be taken into account when analyzing future results, as they reflect specific aspects of each modality. Therefore, this study seeks not only to compare the forms of teaching, but also to understand how these approaches impact on the acquisition of knowledge and the development of leadership skills. The results could contribute to future research and educational disciplines, enabling adaptations that combine the advantages of both modalities.

Among the possible limitations of the study, it is highlighted that as it is a specific population, the sample may undergo changes due to students withdrawing from participation in the collection period, making it necessary to recalculate the sample.

## Disclosure of study results

The present study is part of an academic Master's degree. The dissertation and its results will be presented to the scientific committee of the Universidade Federal do Rio Grande do Norte – UFRN.

## Changes in the study

Any necessary changes to the protocol will be effectively communicated and modified in the relevant parties (trial registry, funding agency and journal). Any questions about the study will be duly answered by the researchers during the initial evaluation period and during the study period.

## Study situation

This study is currently being prepared for participant recruitment.

## Supporting information

**S1 Fig 2. CONSORT 2010 diagram flow.**
(TIF)

**S2 Fig 3. Flowchart of systematization of data collection steps.**
(TIFF)

**S3. Questionnaire.**
(DOCX)

**S4. SPIRIT checklist.**
(DOCX)

**S5. Clinical Trial Registry. REBEC.**
(PDF)

## Author contributions

**Conceptualization:** Jucielly Ferreira da Fonseca, Laura Lima Souza, Daniele Vieira Dantas.

**Formal analysis:** Silmara de Oliveira Silva, Rodrigo Assis Neves Dantas, Daniele Vieira Dantas.

**Investigation:** Daniele Vieira Dantas.

**Methodology:** Jucielly Ferreira da Fonseca, Silmara de Oliveira Silva, Francisco de Cássio de Oliveira Mendes, Rodrigo Assis Neves Dantas, Daniele Vieira Dantas.

**Project administration:** Daniele Vieira Dantas.

**Resources:** Louise Constancia de Melo Alves Silva.

**Supervision:** Silmara de Oliveira Silva, Louise Constancia de Melo Alves Silva, Daniele Vieira Dantas.

**Validation:** Jucielly Ferreira da Fonseca, Louise Constancia de Melo Alves Silva, Francisco de Cássio de Oliveira Mendes, Rodrigo Assis Neves Dantas, Daniele Vieira Dantas.

**Visualization:** Jucielly Ferreira da Fonseca, Francisco de Cássio de Oliveira Mendes.

**Writing – original draft:** Jucielly Ferreira da Fonseca, Louise Constancia de Melo Alves Silva, Roberta Paolli de Paiva Oliveira Arruda Camara, Laura Lima Souza.

**Writing – review & editing:** Jucielly Ferreira da Fonseca, Roberta Paolli de Paiva Oliveira Arruda Camara, Laura Lima Souza, Rodrigo Assis Neves Dantas, Daniele Vieira Dantas.

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
