## [Decision Letter · Decision Letter 0]

21 Aug 2024

PONE-D-24-20769Effect of online module on nursing students’ knowledge acquisition about leadership: A randomized clinical trial protocolPLOS ONE

Dear Dr. Dantas,

Thank you for submitting your manuscript to PLOS ONE. After careful consideration, we feel that it has merit but does not fully meet PLOS ONE’s publication criteria as it currently stands. Therefore, we invite you to submit a revised version of the manuscript that addresses the points raised during the review process.

**- ** The results of this randomized clinical trial will have a positive impact on the participants’ knowledge acquisition, as well as enabling actions/interventions to be developed which promote more effective learning about Leadership in the training context, in turn providing students with greater knowledge and increased confidence in the subject, training nurses capable of promoting an efficient and collaborative work environment, optimizing the results achieved in patient care and contributing to the success of the healthcare institution.

Article accepted with few small corrections mentioned as below-

Author writes the rebuttal against reviewers comments point wise to justify the significance and relevance of study.The method, experimental, statistical and discussion part improved with justified references to understand the study more better way.The significance and future prospect of the study should be emphasize more on clinical benefit.

**Acceptance- Yes**

**Decision- Major revision**

We look forward to receiving your revised manuscript.

Kind regards,

Anshuman Mishra, PhD

Academic Editor

PLOS ONE

Journal Requirements:

4. Thank you for submitting your clinical trial to PLOS ONE and for providing the name of the registry and the registration number. The information in the registry entry suggests that your trial was registered after patient recruitment began. PLOS ONE strongly encourages authors to register all trials before recruiting the first participant in a study.

1) your reasons for your delay in registering this study (after enrolment of participants started);

2) confirmation that all related trials are registered by stating: “The authors confirm that all ongoing and related trials for this drug/intervention are registered”.

5. Please ensure that you refer to Figure 2 and 3 in your text as, if accepted, production will need this reference to link the reader to the figure.

Additional Editor Comments:

Title-Effect of online module on nursing students’ knowledge acquisition about leadership: A randomized clinical trial protocol

Summary of Article- Study by Jucielly Ferreira da Fonseca, et al, proposes to evaluate the impact of an online educational module focused on leadership for Nursing students, recognizing the importance of training future professionals from the beginning of their careers. 

Comments- The results of this randomized clinical trial will have a positive impact on the participants’ knowledge acquisition, as well as enabling actions/interventions to be developed which promote more effective learning about Leadership in the training context, in turn providing students with greater knowledge and increased confidence in the subject, training nurses capable of promoting an efficient and collaborative work environment, optimizing the results achieved in patient care and contributing to the success of the healthcare institution.

Article accepted with few small corrections mentioned as below-

1. Author writes the rebuttal against reviewers comments point wise to justify the significance and relevance of study.

2. The method, experimental, statistical and discussion part improved with justified references to understand the study more better way.

3. The significance and future prospect of the study should be emphasize more on clinical benefit.

Reviewers' comments:

Reviewer's Responses to Questions

**Comments to the Author**

1. Does the manuscript provide a valid rationale for the proposed study, with clearly identified and justified research questions?

Reviewer #1: Yes

Reviewer #2: Yes

2. Is the protocol technically sound and planned in a manner that will lead to a meaningful outcome and allow testing the stated hypotheses?

Reviewer #1: No

Reviewer #2: Yes

3. Is the methodology feasible and described in sufficient detail to allow the work to be replicable?

Reviewer #1: No

Reviewer #2: Yes

4. Have the authors described where all data underlying the findings will be made available when the study is complete?

Reviewer #1: Yes

Reviewer #2: No

5. Is the manuscript presented in an intelligible fashion and written in standard English?

Reviewer #1: No

Reviewer #2: Yes

6. Review Comments to the Author

You may also provide optional suggestions and comments to authors that they might find helpful in planning their study.

Reviewer #1: Comment from the review of Effect of online module on nursing students’ knowledge acquisition about leadership: A randomized clinical trial protocol- PONE-D-24-20769

This study is a protocol for a 2-arm RCT that seek to evaluate the effectiveness of an online Nursing leadership module on the knowledge on leadership of participants who are nursing student in the sixth semester. The study proposed to use two treatment groups: a control group assigned to traditional teaching of nursing leadership and an experiential group assigned to an online nursing leadership module.

The concept is to equip participant with needed leadership skills to help them make good choices in patient care. However, the authors did not report hypotheses to define the concept as well as a clear methodology. This undermines the whole proposal.

The study is timely in the era where the Nursing profession is highly visible with some nurses holding higher offices in various universities and even in the political front across the globe.

However, few concerns were noted during the review, I believe when addressed it will better shape the manuscript.

• The title clinical trial is not appropriate, the setting of the trial is at school and not a clinical site or not involving patients or client- Unless there is some information I couldn’t access in the manuscript.

• Sample size

The sample size is not clear (140,141). Kindly restate it clearly

• Recruitment / Sampling strategy

 the name of the randomised technique should be indicated ie. simple, systematic….? (122)

 The recruitment and allocation is not so clear and it inconsistent under each section it appeared (153-157&172 to 175). Restructure and ensure it flows for easy comprehension.

 with regards to the statement “interested students must register using a form Google forms” (153). what are the students registering for? course, to participate in the study? The students should also register the course in the SIGAA before recruitment into the study (150, 175 &187). clarification of these registrations is important.

 The point at which participants are to be recruited and when they will sign the consent form is inconsistent across subsections (170 and 181) . Also how will the authors divide the two study groups? what strategy will they adopt (156)

• Duration of intervention

 The duration for the intervention is not clearly stated, if the 4 hours is the duration for the intervention as indication in line 200 and 206.Will the intervention take place for only four hours or will it be four hours per week?, or will the it cover for the whole semester ? . what about the workload of 30 hour assigned to the course (150), Authors should reconcile this for clarity?

• Intervention

 You need to indicate how you will deal with confounding factors.

 The allocation of the intervention to the two treatment groups should be well defined from the beginning of the methodology, so that readers can know what exactly you are comparing.

 The proposal should show the differences between the group. ie CG=face to face instruction with traditional time of 4 hours…..? whiles EG= online teaching with flexibility of 5days with added component to the subject matter or teaching methodology such as clinical cases?.

 it is also not clear the mode of teaching researchers will employ for the online class, will it be real-time or recorded lecture? (206 - 208). Please indicate.

• Outcome measurement

 How often will the authors measure the outcome parameters? is it after each lesson? This is not clear in the manuscript (267,272),

 What is to be measured for primary outcome is not clear as well as the tool they will use to measure it? (268-270).

General Comments

1. Authors should clearly state the commonality and differences in the intervention given to in the study groups. ie. clinical case should be clearly defined

2. Randomising technique should be stated and process well defined

3. data collection tool under 219–will be google forms for all, while face to face group will be evaluated post intervention using printed questionnaire (201)

4. Sub heading are saying different things and mostly with additional information that are not relevant preventing the flow of ideas

5. I suggest the definition of terms should be included in the manuscript

6. psychometric properties of the assessment tools should be included in the manuscript to indicate the credibility of the tools.

7. Authors should add more information to the ethical consideration.

8. This study is a randomised control trial, hence the methodology should be more robust and RCT standards be strictly adhered to. Otherwise the authenticity of results will be undermined.

9. no research questions or hypothesis

Reviewer #2: Thank you for the opportunity to review this manuscript. While I find it promising I felt there were a few areas that could be more developed.

Abstract

1. Should include a brief mention of the intended or expected sample size or demographic information to provide context on the study's scope.

Materials and Methods

1. The description of the data collection instruments and the pre- and post-tests could be more detailed, particularly regarding the content of the clinical case and the questionnaire used.

2. It should be more clear how the outcomes are defined, measured, and treated and how the questionnaire and clinical case specifically measure team engagement, organizational culture, conflict management and feedback.

3. The explanation of the experimental intervention (online module) and control intervention (traditional lecture) could benefit from additional details about the specific content and structure.

Blinding

1. The section on data collection could provide more information on how data integrity will be maintained, particularly regarding the handling of digital data from Google Forms and Classroom.

2. While the blinding procedures are detailed, there could be a clearer explanation of how this will be maintained throughout the study, especially concerning post-test administration.

Data Analysis

1. This section could benefit from more specific information about the statistical tests that will be used to compare outcomes between the control and experimental groups.

2. It should specify how missing data will be handled, as this is a common issue in clinical trials.

Discussion

1. While the discussion is generally well-rounded, it could delve deeper into potential challenges or limitations of the study, such as varying levels of student engagement with the online module or differences in baseline knowledge levels

Overall

1. There is a minor inconsistency regarding the semester of students involved (sixth vs. seventh semester), which should be clarified. The abstract reads seventh whereas the methods section reads sixth.

7. PLOS authors have the option to publish the peer review history of their article (what does this mean? ). If published, this will include your full peer review and any attached files.

**Do you want your identity to be public for this peer review?** For information about this choice, including consent withdrawal, please see our Privacy Policy .

Reviewer #1: No

Reviewer #2: No

---

## [Author Response · Author response to Decision Letter 0]

16 Sep 2024

Dear Editor and Reviewers,

Thank you for the suggestions made in the manuscript. We are happy to know that the manuscript was reviewed with great care and attention. We make changes according to the requests of each reviewer and editor. We are available for further clarification. The changes were highlighted in the text of the manuscript in red. Furthermore, the answers in the response letter are in red. Follow the changes made:

- The title clinical trial is not appropriate, the setting of the trial is at school and not a clinical site or not involving patients or client- Unless there is some information I couldn’t access in the manuscript.

Response: Effect of the online module on Leadership on the knowledge acquisition of nursing students: protocol of a randomized controlled trial

- Sample size: The sample size is not clear (140,141). Kindly restate it clearly.

Response: The sample will be for convenience and will include only one class, composed of an average of 40 participants, with the participants randomly divided between the control group and the experimental group..

- Recruitment / Sampling strategy: the name of the randomised technique should be indicated ie. simple, systematic….? (122)

The recruitment and allocation is not so clear and it inconsistent under each section it appeared (153-157&172 to 175). Restructure and ensure it flows for easy comprehension.

With regards to the statement “interested students must register using a form Google forms” (153). what are the students registering for? course, to participate in the study? The students should also register the course in the SIGAA before recruitment into the study (150, 175 &187). clarification of these registrations is important.

The point at which participants are to be recruited and when they will sign the consent form is inconsistent across subsections (170 and 181) . Also how will the authors divide the two study groups? what strategy will they adopt (156)

Response: The recruitment of participants will be conducted in an organized and transparent manner. First, an invitation will be extended to students in the sixth semester of the Nursing undergraduate program at the Federal University of Rio Grande do Norte through the class's WhatsApp group. This invitation will encourage them to enroll in an extension course on leadership.

Interested students must fill out a registration form provided via the Google Forms platform. This form will collect initial participant information and confirm their interest in joining the study. Regarding consent, participants will sign the Informed Consent Form (ICF) at the time of course registration, that is, before the start of activities and before recruitment for the study.

After registration, a Google Meet call will be scheduled with all the registered students. During this call, a pre-test will be administered to assess the participants' prior knowledge on the topic of leadership. Recruitment will be completed with the confirmation of participation from students who sign the consent form and complete the pre-test. This process will ensure that all participants are properly informed and have given their consent to take part in the study.

To ensure random allocation to the groups, simple randomization will be applied using the website www.random.org. This tool will perform the drawing completely randomly, without interference from the researcher, ensuring that all participants have an equal chance of being assigned to the control or experimental group.

Participants in the control group will attend a traditional lecture, held in person at the Department of Nursing. The class, taught by the researcher, will last four hours in the afternoon and cover the same topics that will be addressed in the online educational module. Although the lecture itself is only four hours long, it will be part of a course that, in total, will be considered as having 30 hours, since it includes the time from registration to the completion of activities, for both groups. After the lecture, control group participants will take the post-test.

Meanwhile, participants in the experimental group will take part in an online educational module with recorded lessons hosted on the Google Classroom virtual learning environment. Participants will have a five-day period to access and complete the module content, with a total workload of 30 hours, from registration to completion. The time of access to the platform will take into account the students' enrollment in the platform until the post-test is completed.

It is important to emphasize that the allocation of participants will be conducted in a single-blind manner, thus ensuring the impartiality of the process. Additionally, the assignment of participants will be kept confidential to avoid any external interference or bias on the part of the researcher or others.

- Duration of intervention: The duration for the intervention is not clearly stated, if the 4 hours is the duration for the intervention as indication in line 200 and 206.Will the intervention take place for only four hours or will it be four hours per week?, or will the it cover for the whole semester ? . what about the workload of 30 hour assigned to the course (150), Authors should reconcile this for clarity?

Response: The sample will consist of approximately 40 students who are part of the sixth semester and are regularly enrolled in the first semester of 2024. Throughout their participation, both online and remotely, participants will exclusively use their own mobile phones and internet access to complete the registration and pre-test.

The pre-test will be administered via Google Forms, and no prior support material will be provided. Participants will have 20 minutes to complete the pre-test, and the researcher will use a Google Meet room as a virtual space to host all participants, as everyone must complete the pre-test synchronously within the 20-minute time limit. The virtual platform's chat will be used to signal when participants have finished. All participants must complete the test within the estimated time.

Randomization will be conducted only with participants who complete the pre-test and meet the eligibility criteria. These students will be randomized into two groups: the control group and the experimental group. After randomization, participants will be informed via WhatsApp and email which group they belong to—whether they will attend the course in person at the Department of Nursing or participate in the online educational module. At no point will it be disclosed to the participants which group they belong to.

Participants in the experimental group will be enrolled in a Google Classroom course by the researcher, and instructions will be sent via email and WhatsApp. The online educational module, composed of interactive recorded lessons, will have a four-hour duration, supplemented by additional content such as digital materials, quizzes, and forums. Participants will have flexibility in terms of accessing the lessons, with no geographical limitations.

The control group will participate in an expository-dialogued lecture, held in person at the Department of Nursing at the Federal University of Rio Grande do Norte. Visual materials, such as PowerPoint presentations, will be used during the lecture. The topics covered will be the same as those addressed in the online module, with the same depth and scope.

The experimental group will need to attend the Department of Nursing at UFRN in person. After the control group's activity is completed, experimental group participants will also be present. Following this, the post-test will be initiated for both groups, conducted online through Google Forms. Participants will use their own electronic devices, accessing UFRN's internet. They will be given 20 minutes to complete the post-test. After its completion, all participants will receive a course participation certificate and be dismissed.

Therefore, the pre-test will be conducted synchronously and remotely, using Google Forms, with participants gathered in a Google Meet room. After the pre-test is completed, participants will be allocated and randomized into groups. The post-test will be conducted after all scheduled activities are completed, and this will be done in person, using Google Forms.

The aim of the evaluation is to assess the effectiveness of an online educational module on leadership in the knowledge acquisition of Nursing students.

- Intervention: You need to indicate how you will deal with confounding factors. The allocation of the intervention to the two treatment groups should be well defined from the beginning of the methodology, so that readers can know what exactly you are comparing.The proposal should show the differences between the group. ie CG=face to face instruction with traditional time of 4 hours…..? whiles EG= online teaching with flexibility of 5days with added component to the subject matter or teaching methodology such as clinical cases? it is also not clear the mode of teaching researchers will employ for the online class, will it be real-time or recorded lecture? (206 - 208). Please indicate.

Response: Participants in the experimental group will be enrolled in a Google Classroom course by the researcher, and the instructions will be sent via email and WhatsApp. The online educational module, consisting of interactive recorded lessons, will have a total duration of four hours, supplemented by additional content such as digital materials, quizzes, and forums. The topics covered include team motivation and engagement, organizational culture, conflict management, and constructive feedback. Participants will have flexibility regarding the timing of the lessons and will not face any geographical limitations.

The control group will participate in an expository-dialogued lecture, held in person at the Department of Nursing at the Federal University of Rio Grande do Norte. Visual materials, such as PowerPoint presentations, will be used during the lecture. The topics covered will be the same as those addressed in the online module, with the same depth and scope.

- Outcome measurement: How often will the authors measure the outcome parameters? is it after each lesson? This is not clear in the manuscript (267,272), What is to be measured for primary outcome is not clear as well as the tool they will use to measure it? (268-270).

Response: To evaluate the effectiveness of the module and knowledge acquisition, a data collection tool will be used in the form of a Google Forms questionnaire, which will serve as both the pre-test and post-test. This questionnaire is divided into three parts: a demographic characterization section; a section with a problem-based scenario composed of four objective questions aimed at assessing leadership-related topics such as team motivation and engagement, organizational culture, conflict management, and constructive feedback; and a third section featuring the adapted version of the Student Nurse Leadership Self-Perception Questionnaire (QUAPEEL) by Cardoso, Ramos, and D'Innocenzo (2014). The QUAPEEL consists of questions designed to explore students' knowledge of leadership, allowing for both qualitative and quantitative assessment. It also addresses the skills and attitudes that students believe they possess or need to develop.

Yours sincerely,

The authors.

---

## [Decision Letter · Decision Letter 1]

13 Dec 2024

PONE-D-24-20769R1Effect of the Online Module on Leadership in Knowledge Acquisition among Nursing Students: A Randomized Controlled Study ProtocolPLOS ONE

Dear Dr. Dantas,

Thank you for submitting your manuscript to PLOS ONE. After careful consideration, we feel that it has merit but does not fully meet PLOS ONE’s publication criteria as it currently stands. Therefore, we invite you to submit a revised version of the manuscript that addresses the points raised during the review process.

We look forward to receiving your revised manuscript.

Kind regards,

Anshuman Mishra, PhD

Academic Editor

PLOS ONE

**Additional Editor Comments:**

Dear Authors,

Study- Effect of online modules on nursing students’ knowledge acquisition about leadership: A randomized clinical trial protocol is one of important study for public health, patient care and clinical research perspectives. This study evaluates the effectiveness of Nursing module, healthcare management, treatment groups, etc. However, study lacks several key points and lacks basic information for the standard article (introduction, methodology, discussion, figures, latest references, results, etc.). This undermines the whole review article structure; therefore, I would suggest addressing the reviewers comment point-wise with a clear highlighted version along with a clean version (to understand changes in a better way).

Reveiwers comments are appended below for better understanding.

Reviewers' comments:

Reviewer's Responses to Questions

**Comments to the Author**

1. Does the manuscript provide a valid rationale for the proposed study, with clearly identified and justified research questions?

Reviewer #3: Yes

Reviewer #4: Yes

Reviewer #5: Yes

Reviewer #6: No

Reviewer #7: Yes

Reviewer #8: No

Reviewer #9: No

Reviewer #10: Yes

Reviewer #11: Partly

2. Is the protocol technically sound and planned in a manner that will lead to a meaningful outcome and allow testing the stated hypotheses?

Reviewer #3: Yes

Reviewer #4: Yes

Reviewer #5: Yes

Reviewer #6: No

Reviewer #7: Partly

Reviewer #8: No

Reviewer #9: No

Reviewer #10: Partly

Reviewer #11: Partly

3. Is the methodology feasible and described in sufficient detail to allow the work to be replicable?

Reviewer #3: No

Reviewer #4: Yes

Reviewer #5: Yes

Reviewer #6: Yes

Reviewer #7: Yes

Reviewer #8: No

Reviewer #9: Yes

Reviewer #10: No

Reviewer #11: No

4. Have the authors described where all data underlying the findings will be made available when the study is complete?

Reviewer #3: Yes

Reviewer #4: Yes

Reviewer #5: Yes

Reviewer #6: No

Reviewer #7: No

Reviewer #8: No

Reviewer #9: Yes

Reviewer #10: No

Reviewer #11: Yes

5. Is the manuscript presented in an intelligible fashion and written in standard English?

Reviewer #3: Yes

Reviewer #4: Yes

Reviewer #5: Yes

Reviewer #6: Yes

Reviewer #7: Yes

Reviewer #8: Yes

Reviewer #9: No

Reviewer #10: Yes

Reviewer #11: Yes

6. Review Comments to the Author

You may also provide optional suggestions and comments to authors that they might find helpful in planning their study.

Reviewer #3: The authors have described the methodology in the future tense, making it unclear whether they have performed the study or they are still in the process of performing the study. If they have already conducted the study then the methodology should be described in past tense rather than future tense.

Reviewer #4: 1. Online Module on Leadership in Knowledge Acquisition among Nursing Students is an important research area for the scientific community. Good leadership is important in every profession including nursing as it influences teamwork, decision-making, and the quality of service delivery and patient care. Yes, the manuscript provides a valid rationale for the proposed study, with clearly identified and justified research questions. The research question outlined is expected to address a valid academic problem and contribute to the base of knowledge in the field.

2. Despite the importance of leadership, nursing curricula often prioritize clinical skills over soft skills like leadership and communication. This Protocol is technically sound and planned in a manner that will lead to a meaningful outcome and allow testing of the stated hypotheses. Data analysis will be conducted rigorously, using a randomized controlled trial, a rigorous methodology that will strengthen the reliability of its findings. This method is appropriate and adequate for a study of this nature.

3. Yes, the methodology is feasible and described in sufficient detail to allow the work to be replicable.

4. Yes, the author has described where all data underlying the findings will be made available when the study is complete.

5. Yes, the manuscript is presented in an intelligible fashion and written in standard English.

Additional Comments:

a. Research efforts from past studies in this area should be reviewed.

b. Findings from past research efforts should be integrated to bolster the findings of the current study.

c. Practical implications/applications of the current study's findings should be the focus of the research outcomes. This will serve as the author's contributions to the literature when the study's outcomes are applied to real life issues.

d. Appropriate medical/nursing authorities/regulators should be mentioned in the body and recommendations of the study.

Reviewer #5: The authors have successfully addressed the reviewer's comments. However, the comment posed by the previous reviewer about the outcome measurement that how often will the authors measure the outcome parameters still seems to be unaddressed.

I have also included a few additional minor suggestions for further improvement below.

Title Geographical location: The title of the manuscript effectively conveys the focus of the research, however, to enhance clarity and specificity, it would be beneficial to include the geographical area where the research is going to be conducted. For example, " Effect of the Online Module on Leadership in Knowledge Acquisition among Nursing in Students in Nepal: A Randomized Controlled Study Protocol" would provide readers with a clearer understanding of the study's context (1,2)

Data Analysis: The plan mentions using the Content Validity Index (CVI) and the Kappa test to assess the normality of data distribution. However, these tests are not designed for normality assessment. The CVI is used to measure the validity of items, often in survey research, and the Kappa test measures inter-rater reliability. Instead, normality should be evaluated using tests like the Shapiro-Wilk or Kolmogorov-Smirnov test. This limitation may lead to an inaccurate assumption about data distribution, affecting the choice of statistical tests. (275)

While the authors designate this study as a randomized controlled trial (RCT), there is frequent reference to it as a clinical trial in the abstract. (27,36) To avoid potential confusion, it may be helpful to consistently use the term “randomized controlled trial” to highlight the study’s design rigor and emphasize the randomization process, which is a critical feature in distinguishing it from other types of clinical trials.

Reviewer #6: Thank you for submitting your manuscript, titled "Effect of the Online Module on Leadership in Knowledge Acquisition among Nursing Students: A Randomized Controlled Study Protocol". I appreciate the time and effort you have invested in preparing this work. After careful evaluation, I regret to inform you that your manuscript does not meet the specific standards or requirements of my view. I hope this feedback will help you in further developing your research for publication.

Reviewer #7: I appreciate the authors' efforts to address the suggestions made in the previous review. The revised manuscript shows improvement in clarity and detail. However, some key areas still require further clarification before the manuscript can be considered for publication.

Specific Revisions Requested:

Sample Size: While the authors clarify that the sample will be one class of approximately 40 participants, it would be helpful to justify why a convenience sample is chosen and discuss potential limitations of this approach.

Recruitment and Sampling Strategy: The revised response provides a more detailed description of the recruitment process. However, there are still some inconsistencies that need to be addressed: Clearly state if registration for the Google Forms and SIGAA course are for the same purpose or separate steps.

Ensure consistency in the timing of obtaining informed consent across different sections.

Intervention: The response clarifies the duration and format of the online module. However, it would be beneficial to address potential confounding factors:

Explain how the researchers will minimize interaction between the control and experimental groups to avoid contamination of knowledge.

Further detail the teaching methods employed in the online module (e.g., solely recorded lectures, opportunities for interaction).

Additional Comments: Outcome Measurement: The revised response clarifies the data collection tool and its components. However, consider specifying how often the pre-test and post-test will be administered (e.g., before and after the intervention only).

Overall: I encourage the authors to address the points mentioned above. A more detailed and consistent description of the methodology will strengthen the manuscript and enhance its clarity for readers. Once these revisions are made, I would be happy to re-evaluate the manuscript for publication.

Reviewer #8: The paper lacks depth and does not clearly articulate its contribution to the body of knowledge, aside from a few speculative remarks about the expected outcomes. The scientific methodology employed is questionable. If the outcomes are already presumed before the experiment is conducted, it raises the question: what is the purpose of the experiment?

Firstly, the authors describe the study as a "clinical trial"; however, the content of the paper bears no resemblance to a clinical trial. By definition, a clinical trial is a research study conducted to evaluate the safety, effectiveness, and potential side effects of a medical intervention, such as a new drug, medical device, or treatment method. Clinical trials are critical for advancing medical science and ensuring that new treatments are both effective and safe for patients.

Secondly, the abstract contains duplication; while both versions present the same content, they fail to clearly define the purpose of the research or its actual outcomes. Furthermore, there are ambiguous statements regarding the duration of the experiment—is it 4 hours or 30 hours? There is also ambiguity about the study resources provided to the control group and the experimental group. For instance, will the control group have access to quizzes like the experimental group? How did the authors ensure that neither group was disadvantaged?

Lastly, there is no clear analysis of the results. It is unclear whether the experiment has been conducted or if the study is still in the design phase, as no results are included in the paper. Additionally, there is no conclusion, nor are there any suggestions for future experiments or implications for extending the results—results which, again, have not been described—to a broader cohort.

Suggestions for Improvement:

1. Abstract: Ensure the abstract is concise, not duplicated, and clearly states the purpose, methodology, and expected outcomes of the research.

2. Methodology: Clearly outline the experimental design, including the duration, resources provided to each group, and steps taken to ensure fairness between the control and experimental groups.

3. Result Analysis: Provide a detailed analysis of the experimental results. If the study is in the design phase, clarify this and outline plans for future analysis.

4. Conclusion: Include a conclusion that summarises the findings, suggests directions for future research, and discusses implications for extending the results to a wider cohort.

Reviewer #9: Dear Authors,

Sorry to say that but after reading this paper I can not make a sense who is your participants, Like in various lines you mentioned different cohort:

paze 1 line 27, All seventh period students? what is 7th period ( a course or 7th semester)

paze 2 line 60, All sixth period students? if its 7th or 6th.

page 5 line 137, those enrolled in the sixth period of the course? What is this 6th period or 7th period course.

page 6 line 149, sixth semester of the UFRN Nursing. I am confused if its 1st semester students in 6th period or 7th period or 6th or 7th semester students, so confusing.

Second part when already an reviewer commented that its not a clinical trial but still you mention it at:

paze 1 line 27, clinical trial? it's not a clinical trial.

page 2 line 36 and page 3 line 68, randomized clinical trial

Third thing, the study sample is very low to get any fruitful outcome from this study.

Reviewer #10: this manuscript describes plans for a study to compare learning about and understanding of leadership by nursing students. Although the authors have mostly addressed questions raised by a previous reviewer, I have the following questions, suggestions, and concerns.

One point that should be addressed is that there appear to be numerous differences between the conditions other than modality (in-person vs. online), so what exactly might be concluded from this study other than that this online course differs from this in-person course should be explained

In addition, as noted below in #14, it seems that far more detail should be provided about what data will be coilected and how it will be coded and mapped into whatever response variables will be studied. The data appear to not be just numbers that can be subjected to the ordinary tools of descriptive and inferential statistics. What actually are the variables that will be studied and compared? The absence of this information is why I answered "No" to the question about whether the methodology was described sufficiently to permit the work to be replicable. It seems that a study protocol should be sufficiently detailed so that another researcher could pick it up and implement the study.

1. Although the authors seem to agree with the reviewer of the previous version of this manuscript that "clinical trial" is not an appropriate descriptor for this study, there are still two references to "clinical trial" in the abstract.

2. Within the manuscript, the abstract is duplicated (lines 17-41 and 49-73).

3. Delete the “therefore”s on lines 84 and 86.

4. If it is journal policy, titles of cited articles listed in the reference list should be translated to English.

5. Although the information provided in the cover pages says “N/A” for ethics statement, there is one on lines 116-119.

6. The sentence on lines 124-126 seems to contribute nothing and can probably be deleted. The statement about measurements before and after the intervention might be moved to the end of that paragraph.

7. For the context of this study, it doesn’t really make sense to have separate “Study population” and “Sample size” sections. It’s not really clear what the population is (all nursing students?, all nursing students in Brazil?, all nursing students at this university?), and that probably doesn’t matter. What does matter is that this is a convenience sample that is being divided at random between two groups subject to the constraint that the groups be equal in size. That that constraint is planned should be stated at the same time random assignment is mentioned. However, the constraint is not mentioned in the statement about “allocation” on lines 169-170. If the constraint of equal-sized groups is planned, that has to be stated at around line 170.

8. Will the consent form explain the in-person vs. online difference between the groups? Knowing that the groups differ in modality could influence either rate of agreement or attrition. (This is actually answered on lines 190-192, but should probably also be included where consenting is first mentioned on lines 161-162.

9. It’s not clear what the 30 hours are for the intervention condition. Will participants in that condition be expected to engage with the course for 30 hours (in comparison to the control condition’s four)? Or will the materials be available for 30 hours during the five-day window? Or will they be available for all 120 hours of the five days?

10. The statement on lines 184-186 is somewhat mysterious and should either be deleted or clarified. It’s not clear what single-blinding means in this context. The participants will know in which condition they are participating (but, depending on the answer to comment 8, above, may not know of the existence of the other condition, although it’s hard to imagine how the existence of the two conditions would be revealed by communication between students). If one of the researchers is teaching the lecture course for the control condition, the researcher will know in which condition participants are participating.

The section about Blinding that comes later (lines 248-253) really doesn’t make sense,. It says that participants will be unaware of the condition to which they are allocated, but of course they will be aware of the condition to which they were allocated, especially if, as stated on lines 190-192, they are told about the two conditions as part of the consent process. The only blinding that I can imagine taking place in this study is that coders of any responses that require some sort of coding could be blind to condition.

11. On lines 198-199, what does “both online and remotely” mean? Should this be “both online and in person”?

12. Why, on lines 210-211, does it say “Participants will not be informed of their group

assignment at any point.” when the previous sentence has just said that participants will be informed of their group assignments?

13. Somewhere, probably at around line 230, there should be an explicit statement of how much time will elapse between the online gathering of all participants for the pretest and the in-person gathering of all participants for the posttest.

14. For all the detail that is provided about recruitment and allocation, shockingly little is provided about what will be measured and how. The questionnaire that I understand to provide the “secondary outcomes” data is included (although parts of it are confusing, maybe due to some error in translation—e.g., what does frequency from never to always have to do with understanding statements), the “problem” and questions that provide the information for primary outcomes is not included. As I understand things, each of these includes some open-ended answers, what the questions are and how answers will be coded and mapped into response variables seems to be crucial information for a study protocol.

15, The authors say that data will be made available when a final manuscript is published, but do not say where.

Reviewer #11: 1. First of all, it is randomized control trial and not randomized trial control. please correct it.

2. It is not at all clear to me what methodology will be used by the authors to evaluate the effect of the intervention. is it regression or a simple ttest?

3. Since, the authors are receiving the applications from the students and then they are randomizing. There could be a self selection. Even if you are do randomization after the application are received, how would you generalize your results because your sample is not random anymore. The kind of students who have agreed to be participants have self-selected themselves. Something to think about.

4. Sample size is very small, how would you generalize your results?

5. You have mention about balance test and common support.

7. PLOS authors have the option to publish the peer review history of their article (what does this mean? ). If published, this will include your full peer review and any attached files.

**Do you want your identity to be public for this peer review?** For information about this choice, including consent withdrawal, please see our Privacy Policy .

Reviewer #3: No

Reviewer #4: No

Reviewer #5: **Yes: ** Rima Mishra

Reviewer #6: **Yes: ** Rajeev Kumar Mishra

Reviewer #7: **Yes: ** Rahul Kumar Mishra

Reviewer #8: **Yes: ** Obinna Johnphill

Reviewer #9: **Yes: ** Manish Mishra

Reviewer #10: **Yes: ** Albert F. Smith

Reviewer #11: No

---

## [Author Response · Author response to Decision Letter 1]

24 Jan 2025

Dear Reviewers and Editor of PLOS ONE,

As requested, the authors of the manuscript entitled " Effect of the Online Module on Leadership in Knowledge Acquisition among Nursing Students: A Randomized Controlled Study Protocol" (PONE-D-24-20769R1), have adapted the text respecting the recommendations given by the reviewers.

We would like to point out that a manuscript with the changes highlighted in YELLOW was sent to the journal for consideration and another manuscript was sent in a clean version.

The following changes were made:

Text: As requested by the reviewers.

Reviewer 3

● We would like to clarify that the use of the future tense in the text of the article is due to the fact that the protocol is still under development and preparation for the start of recruitment.

Reviewer 10

● We have deleted the words 'therefore' in lines 84 and 86 of the previous version of the article;

● Page 8, line 183; Page 9, line 215 - We have identified the mistake in the use of the terms 'online and remotely' and have adjusted them to 'online and in person', as suggested;

● Page 12, lines 292-297 - We have clarified in more detail where the data will be available after the research is completed.

Title: As requested by the reviewers.

Reviewer 5

● Regarding reviewer 5's suggestion to include the geographical area in the title of the study, we recognize the relevance of this recommendation. However, we have opted to keep the title as it is, as it is already long and, considering the word limitation for titles, we prefer to preserve it in its original form.

Abstract: As requested by the reviewers.

Reviewer 8

● Page 2, lines 32-37 - The abstract was revised to make it more concise, without duplication, and with a clear statement of the purpose of the study. In addition, the expected results of the research were explicitly included, as suggested.

Reviewers 5, 9, 10 and 11

● Page 1, lines 26-27 - We have revised the use of the term 'clinical trial' in the abstract and throughout the text of the article, replacing it with 'randomized controlled trial', as suggested.

Reviewers 10 and 11

● We identified that the abstract was duplicated in the text and only one version was kept;

Method: As requested by the reviewers.

Reviewer 5

● Pages 13-14, lines 316-318 - We identified a mistake in the use of the 'Content Validity Index' and the Kappa test. We have therefore adjusted the text, replacing these statistical tests with the Shapiro-Wilk test or the Kolmogorov-Smirnov test for assessing normality.

Reviewer 7

● Page 5, lines 123-125 - We have included a more detailed explanation of registration on Google Forms and SIGAA.

● Page 6, lines 137-140 - In addition, we have clarified how researchers will minimize the risk of knowledge contamination between groups.

● Page 5, lines 109-114 - In response to the request of reviewer 7, we have justified the choice of convenience sampling, highlighting the possible limitations associated with this type of sampling;

● Page 9, lines 212- 214 - We specified the frequency of the pre-test and post-test in the text.

Reviewer 8

● Pages 9-11, lines 223-261 - In response to the reviewer's comments, we have clarified the data collection instrument in more detail. In the “Data Collection Instrument” section, we have added all the variables that will be assessed in the instrument for both groups. This ensures that the evaluation of all participants is carried out equally and without any bias.

Reviewer 9

● Line 27; line 105; line 118; line 123; line 182 - We identified a mistake regarding the semester of the students, clarifying that these will be 'sixth semester' students. All errors relating to the use of the terms 'period' and 'seventh semester' have been corrected in the text.

Reviewer 10

● Page 7, lines 167-169 - We have clarified in more detail the sample size and the restriction of equal sizes between the groups, as requested;

● Pages 6-7, lines 152-158; lines 162-165 - In response to the reviewer's question, we provided a more detailed explanation of the course load for the intervention condition in the groups;

● Page 9, lines 212-214 and lines 215-218 - In response to the request, we have added information on the duration of the online meeting for participants during the pre-test and the face-to-face meeting for the post-test;

● Page 10, lines 271-281 - We have clarified the type of blinding used in this research protocol;

● Page 11, lines 226-261 - We have provided more details on the study variables that will be collected;

● Page 11, lines 271-276 - As suggested by the reviewer, we made the differences between in person and online modalities clearer during the consent process.

Reviewer 11

● Page 12, lines 326-333 - We have more clearly specified the statistical tests that will be used to evaluate the effect of the intervention between the groups.

● Page 5, lines 109-114 - In response to reviewer 11's comment, we have clarified the sample size and generalizability of the study in more detail;

● Page 6, lines 147-149 - In response to the reviewer's comment, we have clarified the process of randomizing the students.

Results: As requested by the reviewers.

Reviewer 8

● Page 14, lines 345-248 - We justified the absence of results by explaining that this is a research protocol and therefore only the expected results are available.

● Lines 350-356 - In addition, we provide more details on the primary results of the study.

Discussion: As requested by the reviewers.

Reviewer 10

● Pages 15-17, lines 368-399 - In response to reviewer 10, we have clarified in the discussion of this protocol the differences between in person and online modalities of the leadership course, using references from authors who deal with this subject.

Conclusion: As requested by the reviewers.

Reviewer 8

● We respectfully justify that it is not possible to include a conclusion section summarizing the findings of the study, as the research has not yet started and therefore there are no final results to be presented.

References: As requested by the reviewers.

Reviewer 10

● We revised all the references, translating them into English. In addition, we adjusted the references to Vancouver format, in accordance with PLOS ONE guidelines.

We would like to thank the reviewers for their corrections and for the opportunity to review our article for consideration and possible acceptance by the renowned PLOS ONE journal.

Yours sincerely,

The authors

---

## [Editor Report · Decision Letter 2]

16 Feb 2025

Effect of the Online Module on Leadership in Knowledge Acquisition among Nursing Students: A Randomized Controlled Study Protocol

PONE-D-24-20769R2

Dear Dr. Dantas,

We’re pleased to inform you that your manuscript has been judged scientifically suitable for publication and will be formally accepted for publication once it meets all outstanding technical requirements.

Kind regards,

Anshuman Mishra, PhD

Academic Editor

PLOS ONE

Additional Editor Comments (optional):

Study-Effect of the Online Module on Leadership in Knowledge Acquisition among Nursing Student, by Daniele Vieira Dantas et al., suggest, optimize patient care outcomes and contribute to the overall success of the healthcare institution. The findings from this randomized clinical trial will significantly enhance participants’ knowledge acquisition and help develop strategies or interventions that foster more effective learning about leadership within the training context. This, in turn, will equip students with a deeper understanding of the subject, boosting their confidence and preparing nurses who can cultivate an efficient and collaborative work environment.
---

## [Editor Report · Acceptance letter]

PONE-D-24-20769R2

PLOS ONE

Dear Dr. Dantas,

I'm pleased to inform you that your manuscript has been deemed suitable for publication in PLOS ONE. Congratulations! Your manuscript is now being handed over to our production team.

Kind regards,

on behalf of

Dr. Anshuman Mishra

Academic Editor

PLOS ONE